# Oligometastatic Breast Cancer Patients Treated with High-Dose Chemotherapy and Targeted Radiation: Long-Term Follow-Up of a Phase II Trial

**DOI:** 10.3390/cancers14205000

**Published:** 2022-10-13

**Authors:** Colton Ladbury, Claire Hao, Christopher Ruel, Jason Liu, Scott Glaser, Arya Amini, Jeffrey Wong, Isaac Paz, Lucille Leong, Robert Morgan, Kim Margolin, Stephen Shibata, Paul Frankel, George Somlo, Savita Dandapani

**Affiliations:** 1Department of Radiation Oncology, City of Hope National Medical Center, Duarte, CA 91010, USA; 2Division of Biostatistics, City of Hope National Medical Center, Duarte, CA 91010, USA; 3Department of Surgery, City of Hope National Medical Center, Duarte, CA 91010, USA; 4Department of Hematology and Hematopoietic Cell Transplantation, City of Hope National Medical Center, Duarte, CA 91010, USA; 5Department of Medical Oncology, Saint John’s Cancer Institute, Santa Monica, CA 90404, USA

**Keywords:** oligometastases, radiation, breast cancer, high-dose chemotherapy

## Abstract

**Simple Summary:**

Based on recent clinical trials, radiation is a standard treatment option for limited metastatic sites in metastatic breast cancer, with the potential to improve survival. This is typically given in the form of high-dose radiation called stereotactic body radiotherapy (SBRT). However, SBRT is a newer technology that is not on option for all patients and does not have long-term follow-up. Prior to the widespread implementation of SBRT, we performed a clinical trial utilizing high-dose chemotherapy and standard radiation for metastases in patients with limited metastatic breast cancer. In this research, we analyzed the long-term outcomes of these patients. We found that, despite not using SBRT, radiation provided promising long-term disease control and survival. Therefore, conventional radiation might still be considered if SBRT in not an option for a patient, and our results also help suggest what long-term outcomes of SBRT treatment might look like.

**Abstract:**

Background: Patients with oligometastatic breast cancer (oMBC) may benefit from aggressive local therapy. We sought to assess the effects of consolidative radiation therapy (RT) on outcomes in oMBC patients treated on a prospective phase II trial of high-dose chemotherapy (HDCT). Methods: Between 2005 and 2009, 12 patients with oMBC (≤3 metastatic sites) cancer were treated on protocol. Patients were to receive tandem HDCT supported by hematopoietic cell rescue (HCR). All radiographically identifiable oligometastatic sites received targeted radiation. Results: HDCT was initiated at a median of 6.7 (3.5–12.7) months after diagnosis of oMBC. Hormone receptors (HR) were positive in 91.6% of patients, and HER2 was overexpressed in 25% of patients. Median radiation dose (EQD2) was 41.2 (37.9–48.7) Gy. Median follow-up was 13.1 (6.8–15.1) years for living patients. Ten-year PFS and OS were 33% (95%CI, 10–59%) and 55% (95%CI, 22–79%), respectively. Durable local control of treated lesions was 87.5%. At the last follow up, two patients remained progression free and two more were without evidence of disease following additional salvage treatment. Conclusions: Although modern systemic therapies have obviated the use of HDC, aggressive local therapy warrants further evaluation and fractionated radiotherapy is a viable alternative if SBRT is not available.

## 1. Introduction

Breast cancer is the leading cause of cancer deaths in women worldwide [1]. In the United States, about 4% of women living with breast cancer have metastatic disease (MBC) [2]. Although breast cancer as a whole has favorable survival compared to other cancer types, metastatic disease is still associated with significant mortality, with a 5 year breast-cancer-related survival of 27% between 2015 and 2019 [2]. There has been ongoing interest focusing on patients with limited “oligometastatic” disease (oMBC) [3]; in one set of observations 10 year OS in patients with one metastasis was 17.1% versus 3.2% with five or greater metastases [4]. It has been hypothesized, that oMBC patients might benefit from enhancing systemic treatment with local therapies.

High-dose chemotherapy (HDCT) followed by autologous peripheral blood progenitor cell rescue (PBPC) or hematopoietic cell rescue (HCR), an aggressive form of systemic therapy, has been replaced by improved, less toxic systemic therapies. Meta-analysis of prospective, single-cycle HDCT trials revealed higher complete response rates and improved progression-free survival at early follow-up versus conventional chemotherapy, although no overall survival improvement was seen [5]. Specifically, patients with three or fewer organs involved were felt to benefit most [6,7,8]. Patients treated with tandem cycle HDCT with “debulking” melphalan first, followed by carboplatin, thiotepa, and cyclophosphamide (STAMP-V), were regimens that showed promise [9,10,11]. Further, prior work had demonstrated that, following HDCT, initial failures frequently occurred at previous sites of disease involvement, but local control could be improved with radiation treatment of metastatic sites [12]. Therefore, the role of consolidative radiation in combination with HDCT was previously evaluated in a randomized controlled trial, where it was found to significantly decrease local failure and led to an improvement in PFS and a trend towards an OS benefit, although this trial was not limited to oligometastatic disease and did not necessitate treatment of all visible metastatic sites [13].

We conducted a phase II trial in patients with either locally advanced high-risk (HR) or oMBC. Patients were to receive tandem cycles of HDCT with HCR, and additional targeted radiation to metastatic sites. Five-year results focusing on the PFS, OS, and safety in the HR cohort have previously been published [14]. Herein, we report long-term PFS, OS, and local control (LC) of the oMBC cohort.

## 2. Materials and Methods

### 2.1. Patients

Patients were treated with HDCT, followed by HCR on an IRB-approved prospective Phase II protocol (NCT00182793), following voluntary informed consent, between 2005 and 2009. All procedures performed in studies involving human participants were in accordance with the ethical standards of the institutional and/or national research committee and with the 1964 Helsinki declaration and its later amendments or comparable ethical standards. Informed consent was obtained from all individual participants included in the study. Treatment consisted of melphalan and, upon recovery, STAMP-V as tandem HDCT [15]. Patients with HR LABC and oMBC (with at least partial response to induction systemic therapy) were enrolled between September 2005 and November 2009. The present analysis is limited to patients with oMBC who received targeted radiation.

Patients with oligometastatic disease (≤3 organ sites involved with metastases, regardless of the number of lesions per organ) before induction therapy and ≤3 total residual lesions after systemic induction chemotherapy, with at least partial response, were enrolled. Inclusion criteria included a Karnofsky performance status of ≥80%, age ≤ 65 years, adequate cardiac (left ventricular ejection fraction ≥ 55%), renal (creatinine clearance ≥ 70 mL/min), hepatic (serum aspartate aminotransferase and alanine aminotransferase ≤2 times the upper limit of normal), and pulmonary function, as well as adequate blood counts (neutrophil count of ≥1000/μL and platelet count of ≥100,000/μL). Patients with brain metastases were not eligible. Patients were to undergo apheresis to collect ≥ 4 × 10^6^/kg CD34+ peripheral blood progenitor cells (PBPCs) to qualify [16].

### 2.2. Treatment

#### 2.2.1. HDCT Regimen

All patients completed standard induction systemic therapy for their metastatic breast cancer ≥4 weeks before enrollment. Patients received melphalan 150 mg/m^2^ intravenously over 30 min on day −1. On day 0, 50% of the collected ≥ 4 × 10^6^/kg CD34+ PBPCs were reinfused, and daily 5 μg/kg subcutaneous G-CSF administration was begun. After recovery of marrow function, at a minimum of 5 weeks after cycle 1 of tandem HDCT, patients proceeded with cycle 2, which comprised cyclophosphamide 1.5 g/m^2^/day, carboplatin 200 mg/m^2^/day, and thiotepa 125 mg/m^2^/day given as a continuous intravenous infusion for 96 h on days −7 through −4. (STAMP-V). PBPCs were reinfused on day −2 (12.5% of total) and day 0 (37.5%), with administration of G-CSF starting on day 0.

#### 2.2.2. Additional Therapies

Local-regional radiation therapy, including the primary site and supraclavicular and axillary nodal areas within 6–8 weeks of day 0 of HDCT, was recommended for all patients diagnosed with de novo oMBC. Trastuzumab and antiestrogen therapy (as appropriate) were prescribed.

Patients received targeted radiotherapy to all radiographically identifiable (confirmed by CT and bone scan, and MRI where necessary) metastatic lesions, either before trial enrollment or within 8 weeks of cycle 2 of HDCT. Sites demonstrating complete response to induction therapy were not treated. When spine lesions were treated, one vertebral body above and below the lesion of interest was included in the treatment volume. All radiation was given using a 3D-conformal technique. Because of various dose-fractionation schedules used, an equivalent dose in 2-Gy fractions (EQD2) was calculated, using the linear-quadratic model with an alpha–beta ratio of 4.

### 2.3. Statistical Considerations

Survival outcomes included PFS and OS, calculated from day 0 of cycle 1 of HDCT. Follow-up data were collected up to December 2021. For PFS, the events included death or disease progression, whichever came first. Data for patients who did not experience disease progression and were still alive were censored at the date of last follow-up. For OS, data for patients who were still alive were censored at the date of last follow-up. Local control was defined as lack of progression, based on RECIST criteria, within a treated site’s radiation treatment field, defined as the 90% isodose line. Survival rates were estimated using the Kaplan–Meier method. All statistical analyses were performed using open-source packages in Python 3.8 (Python Software Foundation, Wilmington, DE, USA).

## 3. Results

Twelve patients were enrolled, treated, and evaluable for toxicity and oncologic outcomes. All patients were female and all but two were white (16%). Five patients (42%) were diagnosed with synchronous de novo oMBC, while seven developed metachronous metastases during or after adjuvant treatment. All patients underwent surgical management of the primary tumor (25% lumpectomy and 75% mastectomy) and seven (58%) received adjuvant radiation to the primary following surgery. Only one patient (8%) was postmenopausal at diagnosis. One patient (8%) had triple-negative disease while the remaining were hormone receptor positive. Four patients (33%) also had HER2 amplification. There were two patients who received only endocrine therapy as induction therapy, while 10 patients received chemotherapy as part of systemic induction in order to achieve at least a PR (median number of regimens: 1, range 1–4). Three patients (25%) were unable to receive the second cycle with STAMP-V because of disease progression after melphalan. Nine patients (75%) had bone-only metastases.

All patients received conventional or hypofractionated radiation to affected areas as part of treatment, with nine (75%) receiving radiation to only one metastatic site. Radiated metastatic disease sites included bone (*n* = 13, four in spine), lymph nodes (*n* = 2), and liver (*n* = 1). The median radiation dose was 40 (35–50.4) Gy in a median of 2 (1.8–2.66) Gy fractions. The median EQD2 was 41.2 (37.9–48.7) Gy. The median follow-up was 13.1 (6.8–15.1) years for living patients. Summary patient characteristics are presented in Table 1 and individual patient details are visualized in Table 2. Toxicity from the tandem HDCT regimen has previously been reported [14]. No grade ≥2 radiation-induced toxicity was reported.

The median follow-up time was 13.1 (range: 6.8–15.1) years for living patients and 7.7 (range: 0.7–15.1) years for all patients. In patients with stage IV oligometastatic disease, 5- and 10-year PFS were 33% (95% CI, 10–59%) and 17% (95% CI, 3–41%), respectively (Figure 1A). Median PFS was 3.4 (95% CI, 0.41–6.6) years. Five- and ten-year OS were 75% (95% CI, 41–91%) and 55% (95% CI, 22–78%), respectively (Figure 1B). Median OS was not reached (NR) (95% CI, 2.4-NR) years. Of the 16 metastatic lesions treated, 14 (87.5%) achieved durable local control. The remaining two sites progressed as part of initial relapse, but in each case progressed alongside widespread systemic disease progression. All local failures occurred in bone.

Full patient-level information is detailed in Table 2. All patients who did not complete the tandem transplant are deceased. Two patients remained in remission at 13.1 and 7.2 years. One patient had ER+/PR−/HER2-disease with radiation to a single thoracic spine metastasis to 40 Gy and received ongoing antiestrogen therapy. The other patient had triple-positive disease and had a single liver metastasis radiated to 45 Gy with ongoing antiestrogen and HER2-directed therapy. Two additional patients were successfully salvaged after relapse, one with stereotactic radiosurgery to a brain metastasis, with no further treatment and the other with a pulmonary wedge resection, chemotherapy, antiestrogen therapy, and HER2-directed therapy.

## 4. Discussion

HDCT and HCR have been tested both in the HR and MBC setting, with several phase II and limited numbers of phase III trials showing promise, but with conclusive meta-analyses revealing no overall survival benefits, despite evidence for statistically significant relapse-free survival benefit in high-risk and PFS benefit in patients with metastatic disease [8,17]. Here we report the long-term outcomes of patients with oMBC treated with HDCT, with HCR and targeted radiation to sites of metastatic disease as part of our single-institution phase II trial. To our knowledge, this study represents the longest follow-up after radiation for oMBC in patients treated prospectively in conjunction with HDCT and HCR.

The best historical control for patients with oMBC in general, comes from a nationwide cohort of 3447 patients in the Netherlands diagnosed between 2000 and 2007, with a median follow-up of 15.2 years [4]. The ten-year overall survival for patients with a single metastasis was 17.1%, and 14.9% for patients with three or fewer metastases, versus 3.4% with more than three metastases. Notably, patients with oMBC, defined as three or fewer metastases, experienced improved PFS (hazard ratio (HR): 0.46 (0.29–0.73; *p* = 0.001) and OS (HR: 0.57 (0.36–0.9; *p* = 0.02) when treated with local therapeutic modalities versus those without local therapies. Our median follow-up is the longest of available prospective studies utilizing preplanned radiation for oMBC, particularly in conjunction with HDCT. Our 10-year OS was 55% (95% CI, 22–78%), which far exceeds even the cohort of patients with a single metastasis. This is even more impressive, given that the definition of “oligometastatic disease” used for our trial is more liberal than modern definitions, allowing any amount of lesions as long as they are confined to three organs as long as only three lesions remained after induction therapy, meaning some of the patients treated on our study would not be considered oligometastatic by modern definitions. However, given that our patients did receive aggressive systemic therapy, it is critical to compare them to patients treated under a similar paradigm.

In a prospective trial by Mundt et al., 31 patients were treated with HDCT and HCR, and ten patients underwent conventional radiation to metastatic sites with intent to attain complete response [12]. Of the patients who did not receive radiation, 63.6% failed first, solely in previously involved sites, compared to only 33.3% in the radiation group. A two-year local control of treated lesions was 92.8%. In another prospective study by Carter et al., 74 patients underwent HDCT, and 53 received consolidative conventional radiation to metastatic sites [13]. Sites of first failure were at previously involved sites in 28% of the radiation-treated cohort versus 62% in the no radiation cohort (*p* = 0.02). PFS and OS at 4 years was 31% vs. 21% (*p* = 0.02) and 30% vs. 16% (*p* = 0.2), respectively. While these studies provide valuable insight into the possible beneficial effects of radiation in the context of systemic therapy for oMBC, they lack long-term follow-up.

Data, particularly long-term follow-up on conventional or hypofractionated radiation to metastases in oMBC, are limited. The primary prospective data were a phase II trial reported by Milano et al. that included 48 patients [18]. In this study, the majority (56.3%) received 10-fraction hypofractionated radiotherapy to metastatic sites, while a minority (8.3%) received stereotactic body radiation (SBRT). The study had a median follow-up of 14.3 years for living patients and 4.4 years for all patients. In patients with bone-only metastases (*n* = 12), the 10-year OS and LC were 75% and 100%, respectively, and in patients with non-bone-only metastases (*n* = 36), the 10-year OS and LC were 17% and 73%, respectively. Our study sample size is too small to perform similar subset analyses, but interestingly, bony lesions were the only site of local failure in our study. Further work is therefore needed to know how to best prognosticate and treat patients with oMBC depending on sites of metastatic involvement.

It is important to note that, while our long-term outcomes compare favorably to historical outcomes in MBC, the results were accomplished by now-antiquated techniques in systemic and radiation therapy. While the patients treated on our study received conventional or hypofractionated fractionated radiation, the current standard for radiation to oligometastatic sites is stereotactic body radiotherapy (SBRT), as established by the SABR-COMET trial [19]. This trial included breast, colorectal, lung, prostate, and other cancers, and randomized patients to palliative standard of care treatment with or without SBRT. With a median follow up of 26 months, the median OS trended toward improvement in the SBRT cohort (41 months vs. 28 months, *p* = 0.09), as was progression-free survival defined as lack of local recurrence at the radiated site (12 months vs. 6 months, *p* = 0.0012), but there was no difference in distant disease recurrence rates. SBRT was associated with 5% grade 5 toxicity (vs 0%, *p* = 0.15) and higher rates of related grade 2+ toxicity (29% vs. 9%, *p* = 0.026). Eight-year OS was 27.2% in the SABR arm versus 13.6% in the control arm (*p* = 0.008), while eight-year PFS was 21.3% versus 0.0%, respectively (*p* < 0.001) [20]. The approach has also been evaluated in NR- BR001, wherein patients with breast, lung, or prostate cancer with three to four metastases or two metastases in close proximity were treated with SBRT [21]. With a median follow-up of 22.6 months, the estimated 2-year OS was 57%. No dose-limiting toxicity was observed and grade 3+ AEs occurred in only 19% of patients. This promising data has laid the groundwork for ongoing phase II/III trials (NRG-BR002 and NRG-LU002). Unfortunately, the outcomes of the breast cancer cohorts treated with SBRT are not specified in SABR-COMET (*n* = 13) or NRG-BR001 (*n* = 13).

Prospective data for SBRT in oMBC are available in a Phase II trial by Trovo et al., wherein, at a median of 30 month follow-up, 2-year PFS and OS were 53% and 95%, respectively, with no grade ≥3 toxicity [22]. These early results are consistent with other smaller or retrospective cohorts [23,24,25,26,27,28]. Given what is now known about SBRT based on SABR-COMET, NRG-BR001, and smaller single-arm studies, our long-term results may potentially be improved by combining state-of-the-art systemic therapy in combination with SBRT for oligometastatic disease. Since SBRT is now standard for oligometastatic disease and our trial also includes antiquated systemic options, we are not proposing our trial to be a preferred treatment option in the modern era. However, since radiation to oligometastatic sites is now standard, and SBRT may not always be an option (for example, due to a patient having prior RT to the region in question or receiving treatment at a low-resource center without capability of providing SBRT), we suggest that the more fractionated regimens should still be considered for such patients and may still produce favorable long-term outcomes.

Beyond the efficacy of radiation to oligometastatic sites, another relevant question is patient selection, particularly in the context of modern systemic therapy and risk factors. Modern systemic therapy regimens are different to those used on our trial, including CDK4/6 inhibitors for hormone-positive disease [29], second- and third-generation anti-HER2 therapeutic agents [30], and immunotherapy for triple-negative disease [31]. As systemic therapies improve, it raises the question as to whether they negate the benefit of targeted radiotherapy, although there is also the possible synergy between radiation and immunotherapy generating an abscopal response [32]. Next, in a study of medical oncologists, 86.7% of respondents felt that SBRT could delay growth of metastases and prevent symptoms, and 73% would refer for both symptomatic and asymptomatic metastases [33]. However, the belief did not apply to all patients, as only 63.3% and 50% of respondents were comfortable referring patients with HER2-positive and triple-negative subtypes, respectively. This is likely driven by certain subpopulations (high quantity of metastatic lesions, high metastatic tumor volume, nonbony disease, poor response to systemic therapy, short disease-free interval (<12 months), and triple-negative disease), having increased risk of subsequent distant relapse, and, therefore, potentially a less pronounced benefit of aggressive local treatment [34]. Although lower-risk patients most certainly benefit more from treatment of metastases, it still remains to be determined if the benefit is completely absent in high-risk patients, and, at this time, it is still a reasonable option for all-comers with oMBC if SBRT is felt to be feasible with minimal toxicity. Indeed, three of the four patients treated on our study who are without evidence of disease at the last follow-up have high-risk characteristics.

Our study is limited by its sample size, definition of oligometastatic disease, variability in radiation treatment regimen, use of systemic therapy and radiation methods that are no longer standard in modern oncology practice, and heterogeneity of the patient population. However, given that the early outcomes of our study are comparable to similar patient populations treated with SBRT, it is not unreasonable to extrapolate that, in the long term, consolidative radiation, when given subsequent to state-of-the-art systemic therapy, can be an impactful treatment option for patients with oMBC, and should continue to be an area of focus when seeking to identify optimal patient selection. Furthermore, our results also support the role of conventionally or hypofractionated radiation to oligometastatic sites in patients who may not be able to undergo SBRT due to technological limitations, geographic restrictions, or prior radiation treatment.

## 5. Conclusions

In conclusion, HDCT with HCR and targeted radiation to sites of metastatic disease in oMBC led to favorable long-term PFS and OS in our highly selected patient population, with outcomes comparable to those reported on modern SBRT trials. Although the approaches to systemic therapy and radiation used in this trial are no longer standard, to our knowledge, our study represents one of the longest follow-ups reported on systemic therapy-responsive patients with oMBC treated with prospectively planned targeted radiation in the context of systemic therapy resulting in long-term survival in a subset of patients. Our work also supports fractionated radiotherapy to oligometastases when SBRT is not available. Future work to continue to identify favorable subpopulations of patients with MBC who may benefit from such a treatment paradigm is ongoing and warranted.

## Figures and Tables

**Figure 1 cancers-14-05000-f001:**
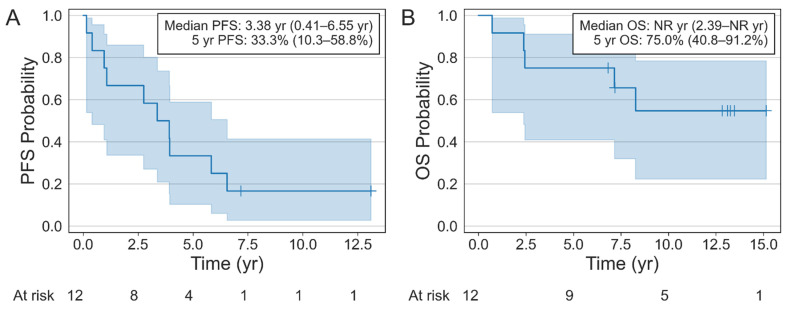
Kaplan–Meier Survival Curves Depicting PFS (**A**) and OS (**B**).

**Table 1 cancers-14-05000-t001:** Patient and treatment characteristics.

Characteristic	*n* (%)
Age (years) (median (range))	44.0 (32–57)
Race/Ethnicity	
Asian	1 (8.3%)
Black	1 (8.3%)
Hispanic White	1 (8.3%)
Non-Hispanic White	9 (75.0%)
KPS	
90	7 (58.3%)
100	5 (41.7%)
Menopausal Status at Diagnosis	
Post	1 (8.3%)
Pre	11 (91.7%)
Stage at Diagnosis	
II	5 (41.7%)
III	1 (8.3%)
IV	5 (41.7%)
Unknown (III or Less)	1 (8.3%)
ER/PR	
−/−	2 (16.7%)
+/−	2 (16.7%)
+/+	8 (66.7%)
HER2	
−	8 (66.7%)
+	4 (33.3%)
Induction therapy *	
Taxane	10 (83.3%)
Anthracycline	4 (33.3%)
Alkylating Agent	5 (41.7%)
Antimetabolite	1 (8.3%)
Antihormone	5 (41.7%)
Other	5 (41.7%)
Time from initial diagnosis to diagnosis of metastasis (months) (median (range))	11.9 (0.0–98.5)
Time from initial diagnosis to protocol treatment (months) (median (range))	20.3 (5.5–103.1)
Time from diagnosis of metastasis to protocol treatment (months) (median (range))	6.7 (3.5–12.7)
Conditioning regimen	
Melphalan alone	3 (25.0%)
Melphalan + STAMP	9 (75.0%)
Radiation dose (EQD2, Gy) (median (range))	41.2 (37.9–48.7)
Number of metastatic sites treated	
1	9 (75.0%)
2	2 (16.7%)
3	1 (8.3%)
Oligometastatic site	
Bone	13 (81.3%)
Liver	1 (6.3%)
Thoracic lymph node	2 (12.5%)

Abbreviations: KPS, Karnofsky performance status; ER, estrogen receptor; PR progesterone receptor; HER2, human epidermal growth factor receptor 2; Gy, Gray; carboplatin, thiotepa, and cyclophosphamide, STAMP; EGD2, equivalent dose in 2 Gy fractions. * In addition to standard adjuvant antiestrogen or HER2-directed therapy.

**Table 2 cancers-14-05000-t002:** Individual patient details.

#	Age	ER/PR/ HER2	Stage at Diagnosis	Transplant Regimen	RT Site	RT Dose (Gy)	RT Fx	Status	Subsequent Treatment *
1	51	−/−/−	II	MEL	T8	35	14	Deceased (7.15 yrs) w/distant progression (5.8 yrs; widespread, no local)	Chemotherapy
2	57	−/−/+	II	MEL	Pretracheal nodes	37.5	15	Deceased (0.7 yrs) w/distant progression (0.2 yrs; widespread, no local)	Chemotherapy, HER2-directed therapy
3	44	+/−/−	IV	MEL + STAMP	T4	40	20	Alive (13.1 yrs) w/o progression	None
4	57	+/+/−	UK (III or less)	MEL	Sternum	40	20	Deceased (2.4 yrs) w/distant progression (0.4 yrs; widespread, no local)	Chemotherapy, antiestrogen therapy
5	43	+/+/+	III	MEL + STAMP	Rib	45	25	Alive (6.8 yrs), w/distant progression (2.8 yrs; lung), NED	Wedge resection, chemotherapy, antiestrogen therapy, HER2-directed therapy
6	32	+/+/−	II	MEL + STAMP	Iliac crest	43.2	24	Alive (12.8 yrs) w/distant progression (6.5 yrs; widespread, no local)	Chemotherapy, palliative radiation, antiestrogen therapy
7	40	+/−/−	II	MEL + STAMP	Sternum, scapula, sacroiliac joint	40	20	Deceased (2.4 yrs) w/distant progression (1.1 yrs; widespread, no local)	Chemotherapy, antiestrogen therapy
8	44	+/+/−	IV	MEL + STAMP	Sternum	50.4	28	Alive (15.1 yrs) w/local and distant progression (1.1 yrs; widespread)	Chemotherapy, antiestrogen therapy
9	55	+/+/+	IV	MEL + STAMP	Liver	45	25	Alive (7.2 yrs) w/o progression	None
10	42	+/+/−	II	MEL + STAMP	T12, L2	35	14	Deceased (8.3 yrs) w/local and distant progression (3.9 yrs; widespread)	Chemotherapy, palliative radiation, antiestrogen therapy
11	42	+/+/+	IV	MEL + STAMP	Contralateral supraclavicular node	50.4	28	Alive (13.5 yrs), w/distant progression (1.0 yrs; brain), NED	Stereotactic radiosurgery
12	48	+/+/−	IV	MEL + STAMP	Bilateral acetabulum	39.9	15	Alive (13.3 yrs) w/distant and local progression (3.9 yrs; widespread)	Chemotherapy, palliative radiation, antiestrogen therapy

Abbreviations: ER, estrogen receptor; PR, progesterone receptor; HER2, human epidermal growth factor receptor 2; MEL, melphalan; carboplatin, thiotepa, and cyclophosphamide, STAMP; RT, radiation therapy; Gy, Gray; Fx, fractions; NED, no evidence of disease. * In addition to standard adjuvant antiestrogen or HER2-directed therapy.

## Data Availability

Research data are stored in an institutional repository and will be shared upon request to the corresponding author.

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
