# Peer review of "Oligometastatic Breast Cancer Patients Treated with High-Dose Chemotherapy and Targeted Radiation: Long-Term Follow-Up of a Phase II Trial"

_cancers, 2022, doi:10.3390/cancers14205000_

Round 1
Reviewer 1 Report
The manuscript deals with a very small (N=12) group of patients with oligometastatic breast cancer who received curative intended systemic therapy including stem cell transplantation plus local therapy of all metastases within a prospective therapy protocol. Relevant conclusions are hardly possible due to the small sample size. Nevertheless, the data are quite interesting because about one fifth of the patients achieved long-term remission over more than 10 years and a survival rate of more than 50% after 15 years was observed including systemic plus local salvage therapies. Of note is the very long follow-up period.
Author Response
Point 1: The manuscript deals with a very small (N=12) group of patients with oligometastatic breast cancer who received curative intended systemic therapy including stem cell transplantation plus local therapy of all metastases within a prospective therapy protocol. Relevant conclusions are hardly possible due to the small sample size. Nevertheless, the data are quite interesting because about one fifth of the patients achieved long-term remission over more than 10 years and a survival rate of more than 50% after 15 years was observed including systemic plus local salvage therapies. Of note is the very long follow-up period.
Response 1: Thank you for your comments.
Reviewer 2 Report
This is a very interesting study addressing a hot topical aspect in oncology: the management of oligometastatic patients. The authors present an original approach, combining chemotherapy and radiotherapy, in a way that is somewhat different from current practice but which, for this very reason, is of great interest due to the alternatives it offers. The study is well conceived, developed and analyzed, presenting the data in a clear, concise and easily understandable manner. The discussion is correct and the conclusions are in agreement with the results observed. The tables and figures provide information that is not redundant with the text.
As suggestions to the authors:
1- It would be advisable for the authors to detail if, during the long follow-up, the patients received additional treatment during progression, what type of treatment, how many times, with what results…
2- Why use an alpha/beta value of 10 when it has long been assumed that the alpha/beta value for breast cancer is more plausibly 3.7? Perhaps it would be advisable for the authors to adapt their EQD2 calculations to a more realistic value.
3.- The authors reference in the discussion, and in reference 19, the Palma et al. article regarding clinical results from SABR-COMET study. The authors should perhaps adapt recently published data of this trial at https://doi.org/10.1016/j.ijrobp.2022.05.004
Author Response
Point 1: It would be advisable for the authors to detail if, during the long follow-up, the patients received additional treatment during progression, what type of treatment, how many times, with what results…
Response 1: Please refer to the last two columns of table 2. In most cases of progression, chemotherapy was given though it was not curative. We do specifically detail in the manuscript two patients who are subsequently NED, following SRS or wedge resection plus chemotherapy.
Point 2: Why use an alpha/beta value of 10 when it has long been assumed that the alpha/beta value for breast cancer is more plausibly 3.7? Perhaps it would be advisable for the authors to adapt their EQD2 calculations to a more realistic value.
Response 2: We agree that an alpha beta of 10 is not applicable here. We have updated the manuscript using an alpha/beta of 4.
Point 3: The authors reference in the discussion, and in reference 19, the Palma et al. article regarding clinical results from SABR-COMET study. The authors should perhaps adapt recently published data of this trial at https://doi.org/10.1016/j.ijrobp.2022.05.004
Response 3: Thank you for bringing our attention to this update. We have updated the manuscript with this additional data in the discussion.
Reviewer 3 Report
Authors have presented a long-term follow-up of a select cohort of 12 patients with oligometastatic breast cancer treated with high dose chemotherapy and stem cell rescue followed by local radiotherapy from a prior phase II single institution study. As recognized by the authors, this treatment plan will not be practiced in the current era of CDK 4/6 inhibitors, immunotherapy and targeted therapy. The HDCT with HCR is a more toxic, expensive and resource consuming strategy which has now been rightfully abandoned. The argument for supporting fractionated or conventional radiotherapy to oligometastatic disease instead of SBRT (in resource poor settings or for patients received prior RT) is reasonable however one cannot extrapolate benefit to patients who will not receive the same type of HDCT/HCR treatment (which will need a tertiary care center). Even the recent phase II NRG-BR002 trial which was a randomized trial failed to demonstrate benefit of local therapy to oligometastatic sites in breast cancer and will not proceed to a phase III component. Essentially, long-term follow-up of this select cohort of patients with oMBC treated with an antiquated treatment regimen does not add or further knowledge about this set of patients.
Author Response
Point 1: Authors have presented a long-term follow-up of a select cohort of 12 patients with oligometastatic breast cancer treated with high dose chemotherapy and stem cell rescue followed by local radiotherapy from a prior phase II single institution study. As recognized by the authors, this treatment plan will not be practiced in the current era of CDK 4/6 inhibitors, immunotherapy and targeted therapy. The HDCT with HCR is a more toxic, expensive and resource consuming strategy which has now been rightfully abandoned. The argument for supporting fractionated or conventional radiotherapy to oligometastatic disease instead of SBRT (in resource poor settings or for patients received prior RT) is reasonable however one cannot extrapolate benefit to patients who will not receive the same type of HDCT/HCR treatment (which will need a tertiary care center). Even the recent phase II NRG-BR002 trial which was a randomized trial failed to demonstrate benefit of local therapy to oligometastatic sites in breast cancer and will not proceed to a phase III component. Essentially, long-term follow-up of this select cohort of patients with oMBC treated with an antiquated treatment regimen does not add or further knowledge about this set of patients.
Response 1: Indeed, NRG-BR002 was a negative trial. However it remains to be seen if there are selected patients who may still benefit, and therefore we still feel this data is worth publishing. The details of the patients treated on the HDCT trial were not extensively discussed in prior publications, so not only does the present study include long-term follow-up, but also adds additional insight into that treatment cohort. We are not advocating for HDCT + conventional RT in this study. We are instead trying to provide long-term follow-up of patients treated with metastasis-directed therapy (not currently available for SBRT trials) and to illustrate that conventional RT can still yield favorable long-term results when SBRT is not feasible for whatever reason.
Reviewer 4 Report
General
Authors provide long-term follow-up on a trial of high dose chemotherapy with standard external beam RT if mets are oligometastatic. Although SBRT is now the standard, the frequency with which we see irradiated sites in general remain under control for the remainder of the course of the disease is fairly high, making the trial and result still pertinent. Also commendable to see longer term follow-up pursued in a disease that often runs a prolonged course.
Abstract
Well written and contains all pertinent data except suggest include timing of RT relative to HDCT.
Length of follow-up useful and control rates of lesions and overall impressive validating this approach in selected patients, accepting contribution of HDCT over standard CT unknown in this scenario.
Introduction
Well written and relevant. Only suggestion, could mention a couple of recent SBRT or other consolidation studies (where such long-term results are as yet lacking) to set the scene further.
Methods
≤3 organ sites involved with metastases, regardless of the number of lesions per organ. This can still be a lot of disease, though ≤3 lesions total after is in keeping with contemporary definitions of oligomets I would say.
Would be interesting to know how many patients started with a low burden v how many finished there due to good response and whether that impacted longer term outcomes. Also how often relapse was at previous disease sites with complete response that were not therefore irradiated to assess whether treating such sites might have helped further.
Results
What was the total number of patients with metastatic disease treated on the HDCT protocol and how many were de novo v relapse? (to get a better idea of how rare or otherwise these patients are)?
Again with table 1 it would be good to see the same demographic data for the whole metastatic cohort, to better judge predictors, ie a second column (accepting the data is present in other papers on this study).
Note the text says one patient was pre-menopausal but table 1 says only one was post-menopausal.
Discussion
As discussed a strength of this paper is the duration of follow-up, not yet possible for SBRT.
An increasing proportion of mBC is de novo as early breast cancer cure rates increase, so it is good to have outcomes for a useful cohort of these patients.
Discussed examples of patients receiving consolidative RT further illustrate the potential benefit of consolidative RT although as not RCTs the patients would likely have been selected.
Regarding SABR-COMET (line 233), text states ‘randomized patients to palliative standard of care treatment versus SBRT’, should be ‘with or without SBRT’, not ‘versus’.
Discussion appropriate including relevance to patients today and limitations.
Author Response
Point 1: Well written and relevant. Only suggestion, could mention a couple of recent SBRT or other consolidation studies (where such long-term results are as yet lacking) to set the scene further.
Response 1: We chose to not focus on SBRT in the introduction since this was not standard at the time the trial began accrual. We do not see a great place to add this so prefer to leave the introduction as is.
Point 2: ≤3 organ sites involved with metastases, regardless of the number of lesions per organ. This can still be a lot of disease, though ≤3 lesions total after is in keeping with contemporary definitions of oligomets I would say.
Response 2: Indeed the definition of oligometastasis has developed since the trial was initially designed. Inclusion would have been different if designed today. We do mention this definition in the limitations section.
Point 3: Would be interesting to know how many patients started with a low burden v how many finished there due to good response and whether that impacted longer term outcomes. Also how often relapse was at previous disease sites with complete response that were not therefore irradiated to assess whether treating such sites might have helped further.
Response 3: Unfortunately information about pre-induction volume was not well captured and is not readily available on retrospective review due to the trial occurring prior to transition to an EMR. Pre chemo imaging is not accessible at this point and the only data available is which organs were initially involved.
Point 4: What was the total number of patients with metastatic disease treated on the HDCT protocol and how many were de novo v relapse? (to get a better idea of how rare or otherwise these patients are)?
Response 4: All patients with metastatic disease were included in the present study. Please refer to Table 1 and 2, which include stage at diagnosis.
Point 5: Again with table 1 it would be good to see the same demographic data for the whole metastatic cohort, to better judge predictors, ie a second column (accepting the data is present in other papers on this study).
Response 5: See response to point 4. All patients with metastatic disease are included in this study. Other patients treated on the HDCT protocol detailed in prior publications did not have metastatic disease and cannot be readily compared to the present study.
Point 6: Note the text says one patient was pre-menopausal but table 1 says only one was post-menopausal.
Response 6: This was an error in the text. It has been fixed to say post-menopausal.
Point 7: Regarding SABR-COMET (line 233), text states ‘randomized patients to palliative standard of care treatment versus SBRT’, should be ‘with or without SBRT’, not ‘versus’.
Response 7: We have updated the manuscript to reflect the correct treatment groups.
Round 2
Reviewer 3 Report
In any trial, there always remains a question whether a select subset may benefit more from investigational therapy even if the overall results are negative. The authors' intentions to provide long-term follow-up of patients treated with metastasis-directed therapy is well received. However cannot extrapolate benefit of metastasis directed therapy for patients with HDCT+HCR to patients not receiving similar therapy. Also, breast clinical trials usually have hundreds of patients and I understand that this was a phase II trial. However, long term follow up of 12 patients with oMBC treated with HDCT+HCR and conventional RT, in my opinion, does not add to existing knowledge or influence clinical practice in anyway.
Author Response
Point 1: In any trial, there always remains a question whether a select subset may benefit more from investigational therapy even if the overall results are negative. The authors' intentions to provide long-term follow-up of patients treated with metastasis-directed therapy is well received. However cannot extrapolate benefit of metastasis directed therapy for patients with HDCT+HCR to patients not receiving similar therapy. Also, breast clinical trials usually have hundreds of patients and I understand that this was a phase II trial. However, long term follow up of 12 patients with oMBC treated with HDCT+HCR and conventional RT, in my opinion, does not add to existing knowledge or influence clinical practice in anyway. Response 1: We acknowledge the reviewer's opinion. We respectfully disagree. Although the specific paradigm used in the study is no longer standard (replaced by better systemic therapies and SBRT). Optimally, these better systemic therapies lead to comparable or better results than HDCT. Although NRG BR002 was negative, due to results of SABR COMET and limited toxicity with SBRT, radiation remains a frequent management option for such patients. In our study, the fact that 4/12 patients are NED at last follow-up is notable and worth publishing. It reasserts that under current treatment paradigms, practitioners can consider conventional RT if SBRT is not an option. The outcomes of our study parallel SABR-COMET's follow-up fairly well, so it is not unreasonable to extrapolate results in the interest of evaluating the modern relevance of conventionally fractionated metastasis directed therapy. We do not disagree that the indication is niche. However, we do not feel that it is not relevant, particularly in the radiation oncology clinic, where scenarios that may warrant metastasis directed therapy, are still prevalent. We do not feel that further revisions will allow us to better address the reviewer's point, as it is directed at the overall study design. We have instead argued in our paper and responses on what we perceive our articles relevance and contribution to be. We have therefore made no further revisions in response to this point.